# HMG-CoA Reductase Inhibitors Suppress Monosodium Urate-Induced NLRP3 Inflammasome Activation through Peroxisome Proliferator-Activated Receptor-γ Activation in THP-1 Cells

**DOI:** 10.3390/ph16040522

**Published:** 2023-03-31

**Authors:** Seong-Kyu Kim, Jung-Yoon Choe, Ji-Won Kim, Ki-Yeun Park

**Affiliations:** 1Division of Rheumatology, Department of Internal Medicine, Catholic University of Daegu School of Medicine, Daegu 42472, Republic of Korea; 2Arthritis and Autoimmunity Research Center, Catholic University of Daegu, Daegu 42472, Republic of Korea

**Keywords:** PPAR-γ, statin, NLRP3 inflammasome, monosodium urate, reactive oxygen species

## Abstract

Peroxisome proliferator-activated receptor γ (PPAR-γ) is thought to negatively regulate NLRP3 inflammasome activation. The aim of this study was to identify the inhibitory effect of 3-hydroxy-3-methylglutaryl coenzyme A (HMG-CoA) reductase inhibitors (statins) on monosodium urate (MSU) crystal-induced NLRP3 inflammasome activation through the regulation of PPAR-γ in THP-1 cells. The expression of PPAR-γ, NLRP3, caspase-1, and interleukin-1β (IL-1β) in human monocytic THP-1 cells transfected with PPAR-γ siRNA or not and stimulated with MSU crystals was assessed using quantitative a real time-polymerase chain reaction and Western blotting. The expression of those markers in THP-1 cells pretreated with statins (atorvastatin, simvastatin, and mevastatin) was also evaluated. Intracellular reactive oxygen species (ROS) were measured using H_2_DCF-DA and flow cytometry analyses. THP-1 cells treated with MSU crystals (0.3 mg/mL) inhibited PARR-γ and increased NLRP3, caspase-1, and IL-1β mRNA and protein expression, and all those changes were significantly reversed by treatment with atorvastatin, simvastatin, or mevastatin. PPAR-γ activity revealed that MSU crystals suppressed PPAR-γ activity, which was markedly augmented by atorvastatin, simvastatin, and mevastatin. Transfecting cells with PPAR-γ siRNA attenuated the inhibitory effect of statins on MSU crystal-mediated NLRP3 inflammasome activation. Statins also significantly reduced the intracellular ROS generation caused by stimulation with MSU crystals. The inhibitory effects of atorvastatin and simvastatin on intracellular ROS generation were reduced in THP-1 cells transfected with PPAR-γ siRNA. This study demonstrates that PPAR-γ is responsible for suppressing MSU-mediated NLRP3 inflammasome activation. The inhibitory effect of statins on MSU-induced NLRP3 inflammasome activation depends on PPAR-γ activity and production and the inhibition of ROS generation.

## 1. Introduction

Gout is a form of chronic inflammatory arthritis caused by the deposition of monosodium urate (MSU) crystals in the intraarticular or periarticular structures of joints, which damages the affected tissues and organs, causing issues such as chronic gouty nephropathy or urolithiasis [1,2]. The mechanisms of acute gouty inflammation have been clarified. Recently, it has been found that the NACHT, LRR, and PYD domain containing protein 3 (NLRP3) inflammasome activation plays a crucial role in the development and progression of uric acid-induced inflammation [3,4,5]. Some medications that reduce the serum uric acid level, such as xanthine oxidase inhibitors and uricosuric agents, and inhibit acute inflammatory response in gout attack are currently the best strategy for managing gout [1,2]. However, novel therapeutic agents are still needed to control the acute inflammatory response in gout.

Peroxisome proliferator-activated receptor-γ (PPAR-γ) is a ligand-dependent transcriptional nuclear regulator of adipocyte differentiation, energy metabolism, and insulin sensitization [6,7]. It exerts an anti-inflammatory effect by suppressing nuclear factor-κB (NF-κB) genes and inhibiting inflammatory cytokines. Recently, some studies found that the PPAR-γ signaling pathway is critically involved in NLRP3 inflammasome activation and consequent inflammatory responses. Comparative gene identification-58 (CGI-58) is a protein responsible for lipid droplet binding and activates adipose triglyceride lipase. Macrophage CGI-58 deficiency promoted high-fat diet-mediated NLRP3 inflammasome activation through the defective maintenance of the PPAR-γ signaling mechanism [8]. In addition, the anti-inflammatory effects of α-linolenic acid metabolites and umbelliferone in sepsis and cerebral/myocardial ischemic animal models were found to be related to the suppression of NLRP3 inflammasome activation via the regulation of PPAR-γ activity and production [9,10,11]. Another study showed a negative association between the PPAR-γ level and caspase-1 expression in human peripheral blood mononuclear cells (PBMCs) from obese subjects [12].

Statins are 3-hydroxy-3-methylglutaryl coenzyme A (HMG-CoA) reductase inhibitors that reduce cholesterol synthesis by limiting the conversion of HMG-CoA to L-mevalonic acid [13,14]. In addition to their lipid-lowering effects, HMG-CoA reductase inhibitors have multiple beneficial pharmacological effects on renal injury, abnormal bone remodeling, and impaired vascular endothelium. The cumulative evidence indicates that statins produce anti-inflammatory and antiatherosclerotic effects by promoting PPAR-γ activation and production [15,16,17]. In addition, HMG-CoA reductase inhibitors suppress NLRP3 inflammasome activation triggered by exposure to various cellular and molecular pathogens and damaged molecules, such as cholesterol or tumor necrosis factor-α [18,19].

Some recent experimental studies have indicated that PPAR-γ is a crucial mediator in regulating MSU crystal-induced NALP3 inflammasome activation in diverse pathological conditions [20,21,22]. Furthermore, clinical data suggest that the PPAR-γ agonist pioglitazone can attenuate the development of uric acid-mediated renal stone or gout [23,24]. However, the anti-inflammatory effects of HMG-CoA reductase inhibitors on uric acid-induced NLRP3 inflammasome activation via the regulation of the PPAR-γ pathway have not been fully elucidated. In this study, we used a human macrophage cell line to investigate whether HMG-CoA reductase inhibitors suppress MSU crystal-induced NLRP3 inflammasome activation by modulating PPAR-γ.

## 2. Results

### 2.1. MSU Crystals Downregulate PPAR-γ Expression in THP-1 Cells

Macrophages treated with MSU crystals (0.3 mg/mL) expressed significantly less PPAR-γ mRNA than untreated cells (Figure 1A). Consistently, PPAR-γ protein expression was dose-dependently reduced in macrophages treated with MSU crystals (Figure 1B). In addition, stimulation with MSU crystals markedly induced the mRNA expression of NLRP3, caspase-1, and IL-1β in a dose-dependent manner (Figure 1A). Consistently, MSU crystals induced NLRP3 protein expression and converted pro-caspase-1 to cleaved caspase-1, which subsequently led to the production of the active inflammatory cytokine IL-1β (Figure 1B).

### 2.2. Statins Increase PPAR-γ Expression and Suppress the NLRP3 Inflammasome Activation Caused by MSU Crystal Stimulation

We assessed whether pretreatment with HMG-CoA reductase inhibitors regulated NLPR3 inflammasome activation and PPAR-γ expression in THP-1 cells stimulated with 0.3 mg/mL MSU crystals. All HMG-CoA reductase inhibitors including atorvastatin, simvastatin, and mevastatin increased PPAR-γ mRNA and protein expression in THP-1 cells treated with MSU crystals (Figure 2A,B). In contrast, NLRP3, caspase-1, and IL-1β mRNA and protein expression induced by MSU crystals was significantly downregulated in cells pretreated with atorvastatin, simvastatin, or mevastatin for 24 h (Figure 2A,B).

We assessed the effect of the statins on PPAR-γ activity in MSU crystal-induced inflammation. Consistent with the inhibitory effect of MSU crystals on PPAR-γ gene and protein expression (Figure 1A,B), we found that MSU crystals (0.3 mg/mL) suppressed PPAR-γ activity (Figure 2C). In contrast, the pretreatment of HMG-CoA reductase inhibitors including atorvastatin, simvastatin, or mevastatin in THP-1 cells treated with MSU crystals significantly increased much more PPAR-γ activity than MSU crystals alone without statins did (Figure 2D). In addition, we found that stimulation with 10 μM of atorvastatin, 10 μM of simvastatin, or 5 μM and 10 μM of mevastatin significantly enhanced PPAR-γ activity (Figure 2E).

### 2.3. PPAR-γ Deficiency Attenuated the Inhibitory Effect of Statins on MSU Crystal-Mediated NLRP3 Inflammasome Activation

We assessed how PPAR-γ affected NLPR3 inflammasome-induced inflammation upon stimulation with MSU crystals. Macrophages transfected with PPAR-γ siRNA markedly induced MSU crystal-induced caspase-1 and IL-1β mRNA expression, compared with cells transfected with NC siRNA but not with NLRP3 mRNA (Figure 3A). Furthermore, in cells transfected with PPAR-γ siRNA, all HMG-CoA reductase inhibitors including atorvastatin, simvastatin, and mevastatin significantly augmented NLPR3, caspase-1, and IL-1β mRNA expression, compared with non-transfected cells. Consistently, NLPR3, caspase-1, and IL-1β protein expression was much higher in macrophages transfected with PPAR-γ siRNA and those treated with statins than in non-transfected macrophages treated with statins (Figure 3B). Those findings suggest that PPAR-γ is required for statin treatment to suppress NLRP3 inflammasome activation.

### 2.4. The Inhibitory Effect of Statins on Intracellular ROS Generation Depends on PPAR-γ in MSU Crystal-Mediated Inflammation

We assessed the antioxidative effects of statins against MSU crystal-induced intracellular ROS generation. The cellular ROS assay using H_2_DCF-DA revealed that MSU crystals induced the production of intracellular ROS in THP-1 cells in a dose-dependent manner, compared with non-treated cells (Figure 4A). Consistently, the flow cytometry measurements detected much more intracellular ROS in cells treated with MSU crystals than in non-treated cells (Figure 4B). The fluorescence intensities of the H_2_DCF-DA probe in THP-1 cells exposed to an HMG-CoA reductase inhibitor (atorvastatin, simvastatin, or mevastatin) were significantly higher than those in cells treated with MSU crystals alone (Figure 4C). Consistently, the flow cytometry analysis showed that statins attenuated intracellular ROS generation compared with cells treated with only MSU crystals (Figure 4D). Interestingly, atorvastatin and simvastatin, but not mevastatin, significantly increased the fluorescence intensities indicating intracellular ROS in macrophages transfected with PPAR-γ siRNA, compared with those of non-transfected cells (Figure 4E). The flow cytometry measurement showed that the cells transfected with PPAR-γ siRNA produced more ROS than non-transfected cells did (Figure 4D,F).

## 3. Discussion

Although the pathogenic mechanisms of gouty arthritis have been identified, novel therapeutic agents to limit uric acid-induced inflammation are still being developed. Recently, some experimental and clinical evidence suggested that PPAR-γ might be an important mediator for controlling uric acid–-induced inflammatory responses or gouty arthritis [21,22,23,24]. Earlier studies found that statins, including atorvastatin, simvastatin, and fluvastatin, induced PPAR-γ activation and then potently suppressed inflammatory responses in macrophage cell lines and human smooth muscle cells [15,16,17]. However, the mechanism through which statins exert their therapeutic effects by regulating the activation and production of PPAR-γ has not been clearly elucidated for uric acid-induced inflammation. In this study, we investigated how statins affect MSU crystal-induced NLRP3 inflammasome activation by regulating PPAR-γ in human macrophage cells. Our main observation is that each statin including atorvastatin, simvastatin, and mevastatin blocked MSU crystal-induced ROS generation by upregulating PPAR-γ, which ultimately suppressed NLRP3 inflammasome activation and the production of IL-1β. In addition, we found that the decrease in PPAR-γ activity and production caused by MSU crystal stimulation is involved in the NLRP3 inflammasome activation that is part of the pathogenesis of gout.

**Figure 5 pharmaceuticals-16-00522-f005:**
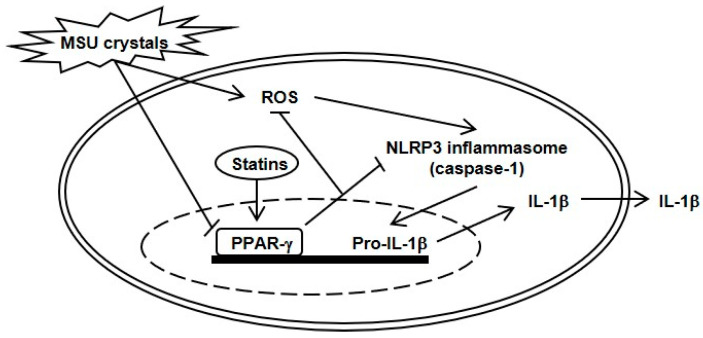
Schematic presentation for the role of statins in MSU crystal-induced NLRP3 inflammasome activation via PPAR-γ. In the process of MSU crystal-induced NLRP3 inflammasome activation and IL-1β production, stimulation with MSU crystals induces suppression of PPAR-γ activity and production. In contrast, THP-1 cells incubated with statins stimulate PPAR-γ activity and production, and then subsequently block ROS generation and NLRP3 inflammasome activation through deactivation of caspase-1. PPAR-γ eventually acts as an inhibitory transcriptional nuclear regulator of uric acid-induced inflammation.

PPAR-γ is generally expressed in white adipocytes and also found in some organs including the kidney, skeletal muscle, and intestine and expressed in diverse inflammatory/immune cells such as macrophages, lymphocytes, and dendritic cells [6,7]. Because of the crucial role of PPAR-γ in regulating glucose and lipid homeostasis, synthetic PPAR-γ agonists such as thiazolidinediones, including rosiglitazone and pioglitazone, have been used to treat hyperglycemia, insulin resistance, and dyslipidemia [7]. Those PPAR-γ ligands, which are called insulin-sensitizing drugs, have been recommended as antidiabetic agents for type 2 diabetes mellitus. In addition to its antidiabetic effect, PPAR-γ has been established to have anti-inflammatory effects by regulating innate immune signaling in macrophages, limiting abnormal cytokine production, or neutralizing inappropriate neutrophil clearance [25]. In particular, the anti-inflammatory mechanism of PPAR-γ upon stimulation with its ligand suppresses inflammatory NF-κB genes by blocking NF-κB signaling and the MAPK pathway activated through TLR2 and TLR4 [26,27]. PPAR-γ is highly expressed at inflammatory lesions in inflammatory arthritis, inflammatory bowel disease, and atherosclerotic plaques and has been considered a novel therapeutic target for those conditions [6]. Given that gout is an inflammatory deforming arthropathy with tophi and associated with insulin resistance, here we investigated the mechanism through which PPAR-γ is involved in the pathogenesis of uric acid-induced inflammation in gout. Interestingly, we found that MSU crystals downregulated PPAR-γ activity and production, suggesting that PPAR-γ could be an important regulator of uric acid-mediated inflammation.

PPAR-γ has been established to have anti-inflammatory effects by blocking the activation of the NLRP3 inflammasome and the production of its proinflammatory cytokines in diverse inflammatory diseases: cerebral ischemia [11], spinal cord injury [28], and obesity [12]. In a rat model of focal cerebral ischemia, PPAR-γ expression was significantly upregulated by umbelliferone, a natural antioxidant and coumarin derivative [11]. Umbelliferone protected against cerebral ischemia reperfusion-induced injury by inhibiting thioredoxin-interactive protein and NLRP3 inflammasome activation and upregulating PPAR-γ activity [11]. Consistently, a spinal cord injury rat model showed increased expression of caspase-1 and proinflammatory cytokines, including IL-1β and IL-6, which was markedly attenuated by treatment with the synthetic PPAR-γ agonist rosiglitazone [28]. In a pre- and post-weight loss surgery analysis of PPAR-γ and caspase-1 levels in PBMCs, the postsurgery patients tended to display reduced caspase-1 and increased PPAR-γ expression compared with the presurgery patients [12]. Hong et al. demonstrated that the PPAR-γ agonist pioglitazone downregulated MSU crystal-induced NALP3 inflammasome activation and subsequently inhibited IL-1β production in HK-2 renal tubular cells [20]. Consistently, we confirmed that PPAR-γ inhibition using siRNA transfection resulted in increased caspse-1 and IL-1β expression upon MSU crystal stimulation. Based on those observations, PPAR-γ might be a crucial transcription factor in NLRP3 inflammasome-mediated inflammation.

Even though the NLRP3 inflammasome is considered to be a key mechanism in explaining the pathogenesis of gout [1,4,5], the clinical relevance of PPAR-γ to gouty arthritis remains elusive. Clinical evidence that synthetic PPAR-γ agonists, such as thiazolidinediones (rosiglitazone and pioglitazone), might have therapeutic value against gout-related inflammatory responses has been proposed. Pioglitazone significantly reduced the incidence of gout in diabetic patients, compared with that in nonpioglitazone users [23]. Another randomized controlled trial demonstrated that pioglitazone decreased the possibility of developing idiopathic uric acid nephrolithiasis by increasing the acidity of urine [24]. Akahoshi et al. found that stimulating human PBMCs with MSU crystals for 2 h or 4 h induced PPAR-γ, suggesting the important role of PPAR-γ in regulating acute gouty inflammation [21]. In contrast, we found that treating THP-1 cells with 0.3 mg/mL MSU crystals for 24 h reduced PPAR-γ activity and also inhibited PPAR-γ gene and protein expression, which suggests that the anti-inflammatory potential of PPAR-γ in the short-lived acute phase of gout is lost when the duration of crystal stimulation becomes longer. Patients need therapeutic agents that can sustain PPAR-γ production and activity throughout the inflammatory response of gout. Yang et al. demonstrated that MSU crystals induced the maturation of caspase-1 and IL-1β, which was markedly inhibited in mouse peritoneal macrophages exposed to the PPAR-γ agonist rosiglitazone [12]. Recently, treatment with rosiglitazone was shown to play a beneficial role in preventing renal inflammation and fibrosis in a rat model of hyperuricemic nephropathy [22]. Although the relationship between PPAR-γ and gout has not been fully elucidated, we assume that evidence can be found to explain how the acute inflammation of gout is mediated through PPAR-γ.

Statins are widely used to treat hypercholesterolemia and reduce cardiovascular events, particularly coronary heart disease [13,14]. Considering the anti-inflammatory mechanism between statins and PPAR-γ, several in vitro studies have found that statin (fluvastatin, pitavastatin, or simvastatin)-induced PPAR-γ activity was significantly attenuated by treatment with a cyclooxygenase-2 (COX-2) inhibitor such as meloxicam or COX-2 siRNA transfection in RAW264.7 cells and human aortic smooth muscle cells (SMCs), but not in human umbilical vein endothelial cells (HUVECs) [15,16]. Given that statins activate COX-2-dependent PPAR-γ expression, the anti-inflammatory effect of statin-induced PPAR-γ was found to involve activating RhoA- and Cdc42-mediated p38 MARK signaling and blocking NF-κB pathway activation. Interestingly, the expression of PPAR-γ differs among different the experimental cells, such as SMCs and HUVECs. PPAR-γ expression was not detected in monocytes, but it was found in human monocyte-derived macrophages [15]. There seems to be crosstalk between statins and PPAR-γ in the process of cholesterol production. Fajas et al. demonstrated that cholesterol depletion for 24 h in undifferentiated 3T3-L1 cells and HepG2 cells markedly induced PPAR-γ protein expression, which is a condition in which adipocyte differentiation and determination factor 1/sterol regulatory element binding protein-1 (ADD-1/SREBP-1) are activated [29]. It was confirmed that the induction of PPAR-γ expression was identified in HepG2 cells cultured with simvastatin. In addition, the induction of PPAR-γ was found to be related to the potential binding of the SREBP transcription factor family to putative E-box motifs in PPAR-γ promoters through a detailed computer-assisted sequence homology analysis. Although this study did not clearly determine the mechanism of statins in PPAR-γ expression, we observed that treating THP-1 cells with statins (atorvastatin, simvastatin, and mevastatin) reversed the attenuation of PPAR-γ activity caused by stimulation with MSU crystals.

ROS generation is considered to be a secondary signaling pathway in the process of NLRP3 inflammasome activation triggered by endogenous and exogenous pathogens [4,5]. Regarding the relationship between HMG-CoA reductase inhibitors and ROS-mediated signaling mechanisms, some studies have indicated that statins show potent antioxidant effects by scavenging ROS or increasing the activity of the antioxidant enzyme thioredoxin [30,31]. In this study, we found that treating macrophages with MSU crystals augmented intracellular ROS generation, which was potently suppressed by the addition of atorvastatin, simvastatin, or mevastatin. Previous evidence suggested that PPAR-γ is a major regulator of ROS generation or ROS-mediated inflammation. The increased intracellular ROS level seen in an ovalbumin-induced murine model of allergic airway disease was markedly suppressed by the PPAR-γ agonists rosiglitazone and pioglitazone [32]. Recently, Hua et al. found that PPAR-γ played a protective role in the advanced glycation end product-induced impairment of coronary artery vasodilation by decreasing ROS production [33]. We also observed that statins decreased intracellular ROS generation by augmenting PPAR-γ activity. Together, the evidence suggests that the anti-inflammatory effects of statins are mechanistically similar to those of PPAR-γ agonists and are thus related to PPAR-γ activation in uric acid-induced inflammation.

There are some limitations in this study. First, we used only one experimental cell line, the human monocytic THP-1 cell. The main goal of this study was to confirm the effect of statin-mediated PPAR-γ ability on NLRP3 inflammasome activation in uric acid-induced inflammation. Considering that gouty inflammation is mainly initiated through NLRP3 inflammasome activation stimulated by MSU crystals in macrophages within joints, the THP-1 cell line could be considered one of the appropriate experimental cell lines to assess the signaling pathway for the NLRP3 inflammasome [34]. Second, we did not verify whether the addition of mevalonate had an inhibitory effect on statin-induced PPAR-γ activation. The reason for not performing an experiment with mevalonate was the precursor of multitudinous metabolites in the mevalonate pathway, which may potentially lead to pleiotropic effects that may control diverse signaling pathways and regulate several cellular functions in macrophages. Ultimately, there may be a limitation in confirming the true effect of mevalonate on statin-induced PPAR-γ activity. Third, we used lipophilic statins such as atorvastatin and simvastatin in this experiment. In addition to lipophilic statins, it could be important to evaluate the ability of lipophilic statins to induce PPAR-γ activation in macrophages compared with that of hydrophilic statins such as rosuvastatin or pravastatin. This is because there are some differences in the membrane permeability and selectivity of target cells such as macrophages or hepatocytes between lipophilic statins and hydrophilic statins [35]. Further studies comparing the efficacy of lipophilic and hydrophilic statins on PPAR-γ activity are needed.

## 4. Materials and Methods

### 4.1. Cell Culture

The human monocytic THP-1 cell line has been widely used to evaluate functions and signaling pathways related to target molecules especially including NLRP3 inflammasomes in monocytes and macrophages [34]. In the experiment, THP-1 cells were cultured in RPMI1640 (Gibco, BRL, Grand Island, NY, USA) supplemented with 10% fetal bovine serum (Hyclone, Logan, UT, USA) and 1% antibiotics (100 units/mL of penicillin and 100 μg/mL of streptomycin). Cells were differentiated with 100 nM of phorbol 12-myristate 13-acetate (PMA, Sigma-Aldrich, St. Louis, MO, USA) for 24 h before stimulation. Atorvastatin, simvastatin, and mevastatin were purchased from Sigma-Aldrich.

### 4.2. Measurement of Intracellular Reactive Oxygen Species (ROS)

We seeded 1 × 10^4^ cells into each well of 96-well black-bottom plates and incubated them with PMA for 24 h. Then, the cells were pretreated with the indicated concentrations of atorvastatin, simvastatin, or mevastatin for 24 h and stimulated with MSU crystals (0.3 mg/mL) for 24 h. After MSU treatment, the medium was removed and incubated with 10 μM 2′,7′-dichlorofluorescein diacetate (H_2_DCF-DA, Thermo, Rockford, IL, USA) for 20 min at 37 °C in the dark. The cells were washed with phosphate-buffered saline, and ROS levels were measured at an excitation wavelength of 485 nm and emission wavelength of 535 nm using an ELISA plate reader (BMG Lab Technologies, Offenburg, Germany).

### 4.3. Flow Cytometry Analysis of ROS

Cellular ROS levels were determined using an H_2_DCF-DA cellular ROS detection assay kit (Abcam, Cambridge, MA, USA) according to the manufacturer instructions. Cells (3 × 10^5^) were seeded on 60 mm culture dishes and incubated with PMA for 24 h. Then, the cells were pretreated with the indicated concentrations of atorvastatin, simvastatin, or mevastatin for 24 h before treatment with MSU crystals (0.3 mg/mL) for 24 h. After staining, the cells were harvested and analyzed using a CytoFLEX flow cytometer (Beckman Coulter, Brea, CA, USA).

### 4.4. RNA Isolation and Quantitative Real Time-Polymerase Chain Reaction (RT-PCR)

Cells (6 × 10^5^) were seeded on 60 mm culture dishes, pretreated with the indicated concentrations of atorvastatin, simvastatin, or mevastatin for 24 h, and then stimulated with MSU crystals (0.3 mg/mL) for 24 h. Total mRNA was extracted using the TRIzol reagent (Gibco BRL, Grand Island, NY, USA). Using a Promega reverse transcription system (Promega, Madison, WI, USA), the extracted RNA was reacted for 20 min at 42 °C and 5 min at 99 °C to obtain complementary DNA. The quantitative PCR amplification was performed using a Mini Option TM RT-PCR system (Bio-Rad, Hercules, CA, USA) and the SYBR green master mix (ToYoBo, Tokyo, Japan). The reactions were carried out with initial denaturation at 95 °C for 15 min, followed by 40 cycles at 9 °C for 5 s, 58–63 °C for 30 s, and 72 °C for 15 s. All reactions were run in triplicate, and the relative expression of each gene was analyzed using the 2^−ΔΔCT^ method.

Primers for PPAR-γ, forward 5-AGG CCA TTT TCT CAA ACG AG-3 and reverse 5-CCA TTA CGG AGA GAT CCA CG-3, NLRP3, forward 5-CAC CTG TTG TGC AAT CTG AAG-3 and reverse 5-GCA AGA TCC TGA CAA CAT GC-3, IL-1β, forward 5-CTG TCC TGC GTG TTG AAA GA-3 and reverse 5-TTG GGT AAT TTT TGG GAT CTA C-3, caspase-1, forward 5-GCG AAG CAT ACT TTC AGT TTC-3 and reverse 5-TCT CCT TCA GGA CCT TGT CG-3, and GAPDH, forward 5-GAC ACC CAC TCC TCC ACC TTT-3 and reverse 5-TTG CTG TAG CCA AAT TCG TTG T-3, manufactured by Bionics (Seoul, Republic of Korea), were used in this experiment.

### 4.5. Transfection of siRNA

Cells (2 × 10^4^) were seeded in 24-well plates and transiently transfected with human PPAR-γ siRNA or negative control siRNA (Invitrogen, Waltham, MA, USA) at a final concentration of 50 nM using Lipofectamine RNAiMAX (Invitrogen, Waltham, MA, USA) according to the manufacturer’s instructions. Briefly, siRNA was diluted in Opti-MEM, mixed with 1 μL of the Lipofectamine RNAiMAX transfection reagent diluted in Opti-MEM, and incubated for 10 min at room temperature. After being cultured for 48 h, the cells were pretreated with statins for 24 h, followed by treatment with MSU crystals (0.3 mg/mL) for 24 h. Then, they were harvested for subsequent experiments.

### 4.6. Western Blotting

Cells were pretreated with the indicated concentrations of atorvastatin, simvastatin, or mevastatin for 24 h and stimulated with MSU crystals (0.3 mg/mL) for 24 h. Total proteins were lysed on ice for 15 min in a radio immunoprecipitation assay buffer (Thermo Scientific, Rockford, IL, USA) containing a protease inhibitor (Roche, Diagnostics, Mannheim, Germany). After centrifugation at 15,000 rpm for 10 min at 4 °C, the supernatants were equalized using the Bradford analysis (Bio-Rad, Hercules, CA, USA).

Cell lysates were subjected to 10% SDS-PAGE and transferred to nitrocellulose membranes (Bio-Rad) by electrophoresis. The membranes were blocked in 5% skim milk (BD Bioscience, San Francisco, CA, USA) and probed with appropriate dilutions of primary antibodies. Antibodies to PPAR-γ and caspase-1 (Abcam, Cambridge, MA, USA), NLRP3 (Novus Biologicals, Littleton, CO, USA), IL-1β and anti-cleaved IL-1β (Cell Signaling, Danvers, MA, USA), and β-actin (Santa Cruz Biotechnology, Santa Cruz, CA, USA) were used.

The membranes were then incubated with horseradish peroxidase-conjugated secondary antibodies for 1 h at room temperature. Proteins were detected using an ECL chemiluminescence kit (Thermo). Densitometry was analyzed and quantified using Quantity One software (Bio-Rad).

### 4.7. PPAR-γ Activity Assay

PPAR-γ activity was measured using a PPAR-γ transcription factor assay kit (ab133101, Abcam, Cambridge, MA, USA) according to the manufacturer’s protocol [36]. Briefly, nuclear extracts from the cells were prepared using a nuclear extraction kit (Abcam, Cambridge, MA, USA). Then, the nuclear proteins were added to 96-well plates precoated with a specific double-stranded DNA sequence containing the peroxisome proliferator response element. The plates were incubated at 4 °C overnight and then washed five times with wash buffer. The plates were incubated with the specific primary anti-PPAR-γ antibody at room temperature for 1 h, followed by incubation with the horseradish peroxidase-conjugated secondary antibody. After washing the plates, we added a developing solution and incubated them for 20 min, after which we added the halting solutions. Absorbance was measured at 450 nm using a microplate reader (BMG Lab Technologies, Offenburg, Germany).

### 4.8. Statistical Analysis

Data are presented as the mean ± standard error of the mean. The statistical differences for each gene and intracellular ROS generation were evaluated using the Kruskal–Wallis test followed by Dunn’s test for multiple comparison. Statistical differences in the expression of target genes according to the dose of MSU crystals or statins were verified with same test. A *p*-value of less than 0.05 was considered statistically significant. The statistical analyses were evaluated using SPSS version 19.0 (SPSS Inc., Chicago, IL, USA). The plots generated in this study were made using GraphPad Prism version 5.04 software (GraphPad Software, San Diego, CA, USA).

## 5. Conclusions

In conclusion, this study has shown that PPAR-γ is required to limit uric acid-mediated NLRP3 inflammasome activation and IL-1β production in a line of human macrophages (Figure 5). Statins potently suppressed MSU crystal-induced NLRP3 inflammasome activation by enhancing PPAR-γ activity in human macrophages. Additionally, the intracellular ROS generated by MSU crystals was downregulated by the anti-inflammatory effect of PPAR-γ. Therefore, PPAR-γ could be a therapeutic target for NLRP3 inflammasome-mediated diseases, including gout. Finally, statins play a potent anti-inflammatory role in NLRP3 inflammasome activation by augmenting PPAR-γ production and thereby strengthening its inhibition of ROS generation in gout.

## Figures and Tables

**Figure 1 pharmaceuticals-16-00522-f001:**
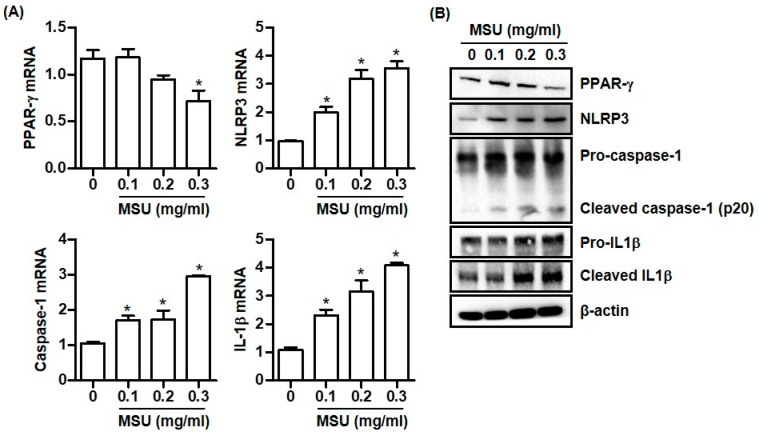
MSU crystals decrease PPAR-γ expression and activate NLRP3 inflammasome. (**A**) PPAR-γ, NLRP3, caspase-1, and IL-1β mRNA expression in THP-1 cells treated with MSU crystals (0.1, 0.2, and 0.3 mg/mL) for 24 h. (**B**) PPAR-γ, NLRP3, caspase-1, and IL-1β protein expression by Western blotting of supernatants and lysates from THP-1 cells treated with MSU crystals for 24 h. * *p* < 0.05 compared to cells treated without MSU crystals using Kruskal–Wallis test followed by Dunn’s multiple comparison test. Values presented as mean ± SEM of three independent experiments. The images are representative of three independent experiments. Abbreviations: PPAR-γ, peroxisome proliferator-activated receptor-γ; NLRP3, The NACHT, LRR, and PYD domains containing protein 3; MSU, monosodium urate; and IL-1β, interleukin-1β.

**Figure 2 pharmaceuticals-16-00522-f002:**
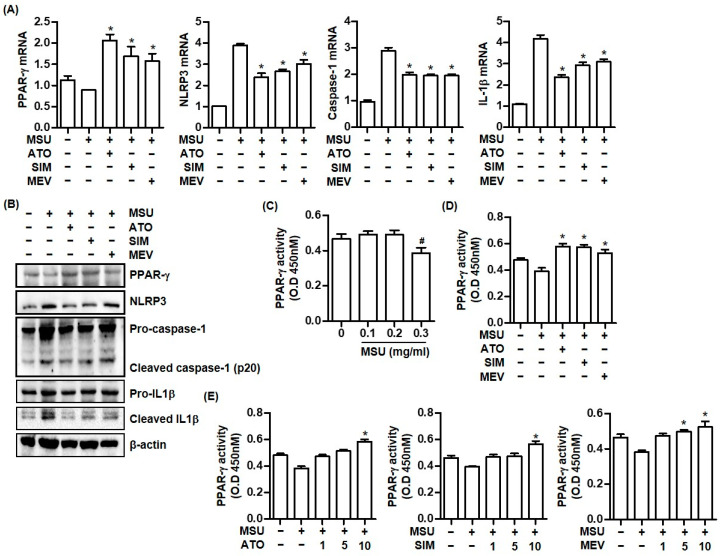
Statins augment PPAR-γ expression and inhibit NLRP3 inflammasome activation in MSU crystal-mediated inflammation. (**A**) PPAR-γ, NLRP3, caspase-1, and IL-1β mRNA expression in THP-1 cells treated with atorvastatin (10 μM), simvastatin (10 μM), or mevastatin (5 μM) for 24 h under stimulation with MSU crystals (0.3 mg/mL) for 24 h. (**B**) PPAR-γ, NLRP3, caspase-1, and IL-1β protein expression by Western blotting of supernatants and lysates from THP-1 cells treated with each statin under stimulation with MSU crystals (0.3 mg/mL) for 24 h. (**C**) PPAR-γ activity measured in nuclear extracts from THP-1 cells treated with MSU crystals (0.1, 0.2, and 0.3 mg/mL). (**D**) Measurement of PPAR-γ activity after addition of atorvastatin (10 μM), simvastatin (10 μM), or mevastatin (5 μM) in THP-1 cells stimulated with MSU crystals. (**E**) Measurement of PPAR-γ activity under stimulation of MSU crystlas (0.3 mg/mL) after pretreatment with multiple dosages of atorvastatin, simavstatin, or mevastatin. *: *p* < 0.05 compared to cells treated with only MSU crystals using Kruskal–Wallis test followed by Dunn’s multiple comparison test; ^#^: *p* < 0.05, compared to cells treated without MSU crystals using Kruskal–Wallis test followed by Dunn’s multiple comparison test. Values presented as mean ± SEM of three independent experiments. Images are representative of three independent experiments. Abbreviations: PPAR-γ, peroxisome proliferator-activated receptor-γ; MSU, monosodium urate; IL-1β, interleukin-1β; ATO, atorvastatin; SIM, simvastatin; and MEV, mevastatin.

**Figure 3 pharmaceuticals-16-00522-f003:**
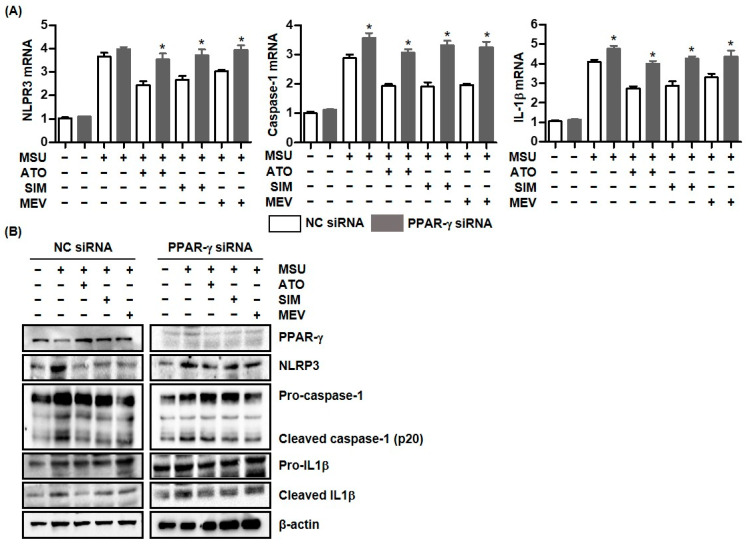
The inhibitory effect of statins on the MSU-induced inflammatory response is attenuated in PPAR-γ deficient cells. (**A**) NLRP3, caspase-1, and IL-1β mRNA expression by addition of statins in THP-1 cells transfected with and without PPAR-γ siRNA under stimulation with MSU crystals (0.3 mg/mL). (**B**) PPAR-γ, NLRP3, caspase-1, and IL-1β protein expression by addition of statins in THP-1 cells transfected with and without PPAR-γ siRNA under stimulation with MSU crystals (0.3 mg/mL). *: *p* < 0.05, compared to cells transfected with NC siRNA using Kruskal–Wallis test followed by Dunn’s multiple comparison test. Values presented as mean ± SEM of three independent experiments. Images are representative of three independent experiments. Abbreviations: PPAR-γ, peroxisome proliferator-activated receptor-γ; MSU, monosodium urate; IL-1β, interleukin-1β; ATO, atorvastatin; SIM, simvastatin; and MEV, mevastatin.

**Figure 4 pharmaceuticals-16-00522-f004:**
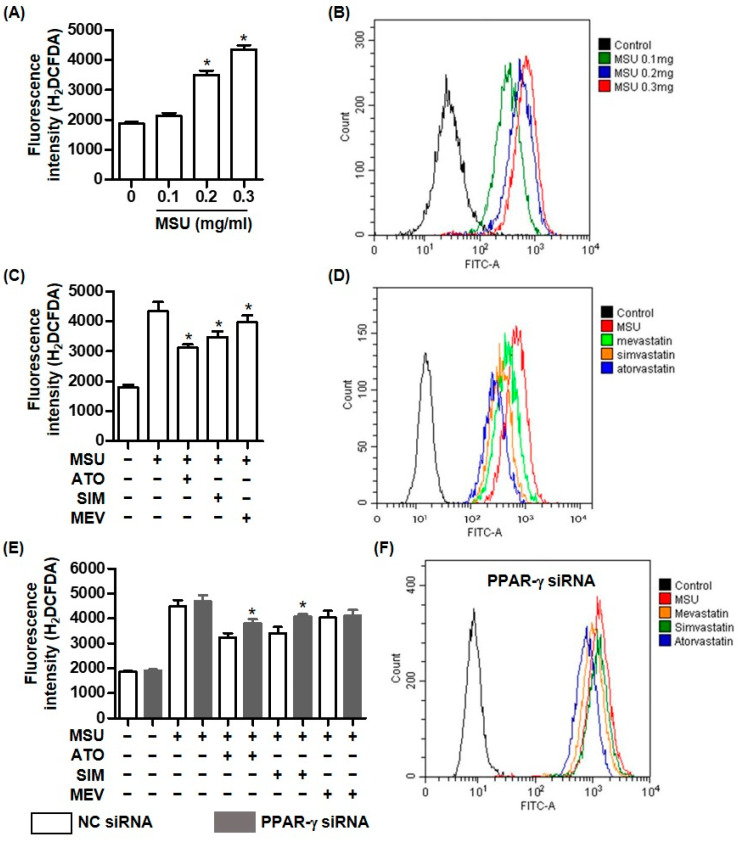
PPAR-γ is required for the inhibitory effect of statins on intracellular ROS generation in MSU crystal-mediated inflammation. (**A**,**B**) Intracellular ROS generation is measured in THP-1 cells treated with MSU crystals (0.1, 0.2, and 0.3 mg/mL) for 24 h using H_2_DCFDA or flow cytometry analysis. (**C**,**D**) Intracellular ROS generation by MSU crystals (0.3 mg/mL) for 24 h measured in THP-1 cells pretreated with statins using H_2_DCFDA or flow cytometry analysis. (**E**) Intracellular ROS generation under stimulation with MSU crystals (0.3 mg/mL) measured in THP-1 cells transfected with and without PPAR-γ siRNA and pretreatment with statins using H_2_DCFDA. (**F**) Intracellular ROS generation under stimulation with MSU crystals (0.3 mg/mL) measured in THP-1 cells transfected with PPAR-γ siRNA and pretreatment with statins using flow cytometry analysis. * *p* < 0.05, compared to cells treated without MSU crystals (**A**), cells treated with only MSU crystals (**C**), and cells transfected with NC siRNA (**E**) using Kruskal–Wallis test followed by Dunn’s multiple comparison test. Values presented as mean ± SEM of three independent experiments. The images are representative of three independent experiments. Abbreviations: MSU, monosodium urate; ATO, atorvastatin; SIM, simvastatin; and MEV, mevastatin.

## Data Availability

Data is contained within the article.

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
