# Peer review of "HMG-CoA Reductase Inhibitors Suppress Monosodium Urate-Induced NLRP3 Inflammasome Activation through Peroxisome Proliferator-Activated Receptor-γ Activation in THP-1 Cells"

_pharmaceuticals, 2023, doi:10.3390/ph16040522_

Round 1
Reviewer 1 Report
In the present study authors report that treatment of THP-1 monocytes with monosodium urate decreases the expression of PARR-gamma and increased NLRP3, caspase-1, and IL-1 mRNA and protein expression, and these changes are reversed by treatment with atorvastatin, simvastatin, or mevastatin. PPAR-gamma siRNA attenuated the protective effect of statins. The results provide the mechanistic background for the possible application of statins in the treatment of gout.
The topic and the results are of interest, however, there are also important concerns to be addressed.
Specific comments:
1) The cell type used in the experiments should be specified in the title.
2) Page 2, line 11, the role of CGI-58 should be explained.
3) Page 2, paragraph 3: effects on glucose metabolism are listed among pleiotropic beneficial effects of statins. However, recent clinical trials and experimental studies indicate that statins may deteriorate glucose metabolism and increase the incidence of type 2 diabetes. Therefore, the effect of statins on glucose could not be considered as beneficial.
4) Page 2, paragraph 4, the phrase: “uric acid–mediated urinary acidosis or gout” should be revised. What is “urinary acidosis”?
5) Section 4.4: primer sequences should be presented. In addition, housekeeping gene used for standardization of RT-qPCR results should be specified.
6) Section 4.7: were the results of PPARgamma activity assay normalized per unit of total PPARg protein? The amount of PPARg bound to DNA could be affected by total amount of PPARgamma in the cell in addition to PPARgamma activity.
7) Statistical analysis: Mann-Whitney U test is designed to compare 2 groups. Because there were more groups in the present study, the other appropriate methods should be used.
8) Were meva- and simvastatin used as the lactone form or the open acid form?
9) The mechanism through which statins activate PPAR-gamma should be discussed.
10) Statins inhibit HMG-CoA reductase which converts HMG-CoA to mevalonate. Supplementation of mevalonate is usually used to verify if the effect of statins results from the inhibition of this enzyme or is attributed to some of-target effects. Was this issue addressed by the authors?
11) Study limitations should be discussed. For example, authors used only one cell type. In addition, statin concentrations used in this study are much higher than observed in statin-treated patients.
Author Response
Dear Editor
Manuscript ID: pharmaceuticals-2290465
Type: Article
Title: HMG-CoA reductase inhibitors suppress monosodium urate-induced NLRP3 inflammasome activation through peroxisome proliferator-activated receptor-gamma activation
Thank for the editor and reviewers of the ‘Pharmaceuticals’ for reviewing our manuscript. We have made some corrections and clarifications in the revised manuscript according to the editor’s or reviewer's comments. You can find out tracing marks for changes in revised manuscript. The changes are summarized below:
In the present study authors report that treatment of THP-1 monocytes with monosodium urate decreases the expression of PARR-gamma and increased NLRP3, caspase-1, and IL-1b mRNA and protein expression, and these changes are reversed by treatment with atorvastatin, simvastatin, or mevastatin. PPAR-gamma siRNA attenuated the protective effect of statins. The results provide the mechanistic background for the possible application of statins in the treatment of gout.
The topic and the results are of interest, however, there are also important concerns to be addressed.
Specific comments:
1) The cell type used in the experiments should be specified in the title.
Answer) Thanks for your comment. The cell type used in the experiment is THP-1 cells, which is added at title of this study. Therefore, the title of this study is changed into “HMG-CoA reductase inhibitors suppress monosodium urate-induced NLRP3 inflammasome activation through peroxisome proliferator-activated receptor-g activation in THP-1 cells”.
2) Page 2, line 11, the role of CGI-58 should be explained.
Answer) Thanks for your kind comment. CGI-58 stands for comparative gene identification-58. It is known as α/β-hydrolase domain-containing 5 and plays a role as the co-activator of adipose triglyceride lipase, which is a key enzyme initiating cytosolic lipid droplet lipolysis. We add a sentence for CGI-58 like this, “Comparative gene identification-58 (CGI-58) is a protein responsible for lipid droplet binding and then activates adipose triglyceride lipase.”.
3) Page 2, paragraph 3: effects on glucose metabolism are listed among pleiotropic beneficial effects of statins. However, recent clinical trials and experimental studies indicate that statins may deteriorate glucose metabolism and increase the incidence of type 2 diabetes. Therefore, the effect of statins on glucose could not be considered as beneficial.
Answer) Thanks for your comment. We also agree with your opinion. In the sentence mentioned by the reviewer, "abnormal glucose metabolism" should be deleted within sentence because it was thought to be an inappropriate description that could cause confusion.
4) Page 2, paragraph 4, the phrase: “uric acid–mediated urinary acidosis or gout” should be revised. What is “urinary acidosis”?
Answer) Thanks for your comment. We recognized that the sentence is wrongly presented. Especially “urinary acidosis” is not appropriated to understand the sentence. Therefore, “urinary acidosis” is revised into “renal stone”.
5) Section 4.4: primer sequences should be presented. In addition, housekeeping gene used for standardization of RT-qPCR results should be specified.
Answer) Thanks for your comments. Thus, we add primers for target genes and a housekeeping gene, as follows; “Primers for PPAR-g: forward 5-AGG CCA TTT TCT CAA ACG AG-3; reverse 5-CCA TTA CGG AGA GAT CCA CG-3; NLRP3: forward 5-CAC CTG TTG TGC AAT CTG AAG-3; reverse 5-GCA AGA TCC TGA CAA CAT GC-3; IL-1b: forward 5-CTG TCC TGC GTG TTG AAA GA-3; reverse 5-TTG GGT AAT TTT TGG GAT CTA C-3; caspase-1: forward 5-GCG AAG CAT ACT TTC AGT TTC-3; reverse 5-TCT CCT TCA GGA CCT TGT CG-3, and GAPDH: forward 5-GAC ACC CAC TCC TCC ACC TTT-3; reverse 5-TTG CTG TAG CCA AAT TCG TTG T-3 manufactured by Bionics (Seoul, Korea) were used in this experiment.”.
6) Section 4.7: were the results of PPARg activity assay normalized per unit of total PPARg protein? The amount of PPARg bound to DNA could be affected by total amount of PPARg in the cell in addition to PPARg activity.
Answer) Thanks for your comment. PPAR-g Transcription Factor Assay kit (ab133101) is a non-radioactive, sensitive method for detecting specific transcription factor DNA binding activity in nuclear extracts. Unfortunately, we did not assess total amount of PPAR-g in the cell. Therefore, this experiment only measured the activity of PPAR-g in nuclear protein. We add new reference (No. 37) used PPAR-g transcription factor assay kit manufactured by Abcam.
Reference No. 37) Shou, X.; Zhou, R.; Zhu, L.; Ren, A.; Wang, L.; Wang, Y.; Zhou, J.; Liu, X.; Wang, B.; Emodin, A. Chinese Herbal Medicine, Inhibits Reoxygenation-Induced Injury in Cultured Human Aortic Endothelial Cells by Regulating the Peroxisome Proliferator-Activated Receptor-γ (PPAR-γ) and Endothelial Nitric Oxide Synthase (eNOS) Signaling Pathway. Med. Sci. Monit. 2018, 24, 643-651.
7) Statistical analysis: Mann-Whitney U test is designed to compare 2 groups. Because there were more groups in the present study, the other appropriate methods should be used.
Answer) Thanks for your valuable comment. We recognize that our statistical analysis was wrongly performed. Therefore, statistical differences among multiple groups were compared using Kruskal-Wallis test with Dunn’s test for multiple comparisons as appropriate. The method for statistical analysis is revised as follows; “Data are presented as the mean ± standard error of the mean. The statistical differences for each gene and intracellular ROS generation were evaluated using the Kruskal-Wallis test followed by Dunn’s test for multiple comparison. Statistical differences in the expression of target genes according to the dose of MSU crystals or statins were verified with same test. A p-value of less than 0.05 was considered statistically significant. The statistical analyses were evaluated using SPSS version 19.0 (SPSS Inc., Chicago, IL, USA). The plots generated in this study were made using GraphPad Prism version 5.04 software (GraphPad Software, San Diego, CA, USA).”.
8) Were meva- and simvastatin used as the lactone form or the open acid form?
Answer) Thanks for your valuable comment. Statins have both acid and lactone forms in vivo. The lactone form plays an inhibitory effect on mitochondrial complex III and modulates proteasome activities, while the acid form suppresses HMG-CoA reductase enzymatic activity and then results in inhibition of cholesterol synthesis. We purchased three kinds of statins used in this experiment from Sigma-Aldrich. However, even in the detailed instructions for statins, we could not confirm whether these chemicals we used was in acid or lactone form. Therefore, considering the effect of inhibiting HMG-CoA reductase in this experiment, it is assumed to be in the acid form.
9) The mechanism through which statins activate PPAR-g should be discussed.
Answer) Thanks for your thoughtful comment. We already described some discussion for the mechanism of statin on PPAR-g at page 9. But it might be insufficient to fully explain of the mechanism. Therefore, we additionally discuss evidence like this, “There seems to be cross-talk between statins and PPAR-g in the process of cholesterol production. Fajas et al. demonstrated that cholesterol depletion during 24 h in undifferentiated 3T3-L1 cells and HepG2 cells markedly induced PPAR-g protein expression, which was a condition in which adipocyte differentiation and determination factor 1/sterol regulatory element binding protein-1 (ADD-1/SREBP-1) are activated (30). It was confirmed that induction of PPAR-g expression was identified in HepG2 cells cultured with simvastatin. In addition, induction of PPAR-g was found to be related to potential binding of the SREBP transcription factor family to putative E-box motifs in PPAR-g promoters through a detailed computer-assisted sequence homology analysis.”.
Reference No. 30) Fajas, L.; Schoonjans, K.; Gelman, L.; Kim, J.B.; Najib, J.; Martin, G.; Fruchart, J.C.; Briggs, M. Spiegelman, B.M.; Auwerx, J. Regulation of peroxisome proliferator-activated receptor gamma expression by adipocyte differentiation and determination factor 1/sterol regulatory element binding protein 1: implications for adipocyte differentiation and metabolism. Mol. Cell. Biol. 1999, 19: 5495-5503.
10) Statins inhibit HMG-CoA reductase which converts HMG-CoA to mevalonate. Supplementation of mevalonate is usually used to verify if the effect of statins results from the inhibition of this enzyme or is attributed to some of-target effects. Was this issue addressed by the authors?
Answer) Thanks for your comment. Your comment seems to be very valuable. However, the reason why we did not confirm the effect of mevalonate administration on statin-induced PPAR-g activation in our study is described in the limitations of the study, as follows; “Second, we did not verify whether addition of mevalonate had an inhibitory effect on statin-induced PPAR-g activation. The reason for not performing experiment with mevalonate is that mevalonate is that the precursor of multitudinous metabolites in the mevalonate pathway, potentially leading to pleiotropic effects that may control diverse signaling pathways and regulate several cellular functions in macrophages. Ultimately, there may be a limitation in confirming the true effect of mevalonate on statin-induced PPAR-g activity.”.
11) Study limitations should be discussed. For example, authors used only one cell type. In addition, statin concentrations used in this study are much higher than observed in statin-treated patients.
Answer) Thanks for your thoughtful comments. We also recognize that there are some limitations in understanding the study results due to the lack of experimental methods or additional verification using drugs involved in signaling pathways for target molecules of this study. We discuss the additional comments, as follows “There are some limitations in this study. First, we used only one experimental cell line, human monocytic THP-1 cell. The main goal of this study was to confirm the effect of statin-mediated PPAR-g ability on NLRP3 inflammasome activation in uric acid-induced inflammation. Considering that gouty inflammation mainly initiates through NLRP3 inflammasome activation stimulated by MSU crystals in macrophages within joints, THP-1 cell line could be considered one of appropriate experimental cell lines to assess signaling pathway for NLRP3 inflammasome (35). Second, we did not verify whether addition of mevalonic acid had an inhibitory effect on statin-induced PPAR-g activation. The reason for not performing experiment with melanovic acid is that mevalonic acid is that the precursor of multitudinous metabolites in the mevalonate pathway, potentially leading to pleiotropic effects that may control diverse signaling pathways and regulate several cellular functions in macrophages. Ultimately, there may be a limitation in confirming the true effect of mevalonic acid on statin-induced PPAR-g activity. Third, we used lipophilic statins such as atorvastatin and simvastatin in this experiment. In addition to lipophilic statins, it could be important to evaluate ability of lipophilic statins to induce PPAR-g activation in macrophages compared with hydrophilic statins such as rosuvastatin or pravastatin. This is because there are some differences in membrane permeability and selectivity of target cells such as macrophages or hepatocytes between lipophilic statins and hydrophilic statins (36). Further studies comparing efficacy of lipophilic and hydrophilic statins on PPAR-g activity are needed.”.
Reference No. 35) Chanput, W.; Mes, J.J.; Wichers, H.J. THP-1 cell line: an in vitro cell model for immune modulation approach. Int. Immunopharmacol. 2014, 23, 37-45.
Reference No. 36) Paumelle, R.; Staels, B. Peroxisome proliferator-activated receptors mediate pleiotropic actions of statins. Circ. Res. 2007, 100, 1394-1395.

Reviewer 2 Report
Current report investigated the inhibitory effect of 3-hydroxy-3-methylglutaryl coenzyme A (HMG-CoA) reductase inhibitors (statins) on monosodium urate (MSU) crystal-induced NLRP3 inflammasome activation through the regulation of
PPAR-γin THP-1 cells. I like to give the following comments.
1. The used HMG-CoA reductase inhibitors did not introduce in clear.
2. Reason(s) to apply the THP-1 cells remained unknown.
3. In Figure 1, it shows Western blotting data but not mRNA data in cells. Additionally, two indicators of caspase-1 and IL-1β must mention in clear in the legends.
4. In Figure 2, measurement of PPAR-γactivity was unknown. Additionally, dose-dependent effect of each inhibitor is required.
5. In Figure 3, THP-1 cells transfected with or without PPAR-γsiRNA need to show in clear. Same as for that in Figure 4.
6. In Figure 5, NLRP3 inflammasome activation through deactivation of caspase-1 that was ignored in the figure. Why?
7. Did antioxidant show same effects as Statins?
8. PPAR-γactivity was measured by an assay kit (Abcam) that needs reference(s) to support.
9. Variations between Statins were not conducted. Why?
Author Response
Dear Editor
Manuscript ID: pharmaceuticals-2290465
Type: Article
Title: HMG-CoA reductase inhibitors suppress monosodium urate-induced NLRP3 inflammasome activation through peroxisome proliferator-activated receptor-gamma activation
Thank for the editor and reviewers of the ‘Pharmaceuticals’ for reviewing our manuscript. We have made some corrections and clarifications in the revised manuscript according to the editor’s or reviewer's comments. You can find out tracing marks for changes in revised manuscript. The changes are summarized below:
Current report investigated the inhibitory effect of 3-hydroxy-3-methylglutaryl coenzyme A (HMG-CoA) reductase inhibitors (statins) on monosodium urate (MSU) crystal-induced NLRP3 inflammasome activation through the regulation of PPAR-γ in THP-1 cells. I like to give the following comments.
- The used HMG-CoA reductase inhibitors did not introduce in clear.
Answer) Thanks for your valuable comment. We also agree to describe detailed information of statins used in this experiment. Instead of introduction, please understand that some of the related descriptions is added at the part of study limitation of discussion, as follows, “Third, we used lipophilic statins such as atorvastatin and simvastatin in this experiment. In addition to lipophilic statins, it could be important to evaluate ability of lipophilic statins to induce PPAR-g activation in macrophages compared with hydrophilic statins such as rosuvastatin or pravastatin. This is because there are some differences in membrane permeability and selectivity of target cells such as macrophages or hepatocytes between lipophilic statins and hydrophilic statins (36). Further studies comparing efficacy of lipophilic and hydrophilic statins on PPAR-g activity are needed.”.
Reference No. 36) Paumelle, R.; Staels, B. Peroxisome proliferator-activated receptors mediate pleiotropic actions of statins. Circ. Res. 2007, 100, 1394-1395.
- Reason(s) to apply the THP-1 cells remained unknown.
Answer) Thanks for your comment. The reason why THP-1 cells were used in this experiment is added in the part of 4.1. Cell culture as follows; “THP-1 cell line has been widely used to evaluate functions and signaling pathways related to target molecules especially such as NLRP3 inflammasomes in monocytes and macrophages (35).”.
Reference No. 35) Chanput, W.; Mes, J.J.; Wichers, H.J. THP-1 cell line: an in vitro cell model for immune modulation approach. Int. Immunopharmacol. 2014, 23, 37-45.
- In Figure 1, it shows Western blotting data but not mRNA data in cells. Additionally, two indicators of caspase-1 and IL-1β must mention in clear in the legends.
Answer) Thanks for your valuable comment. We now recognize that the detailed description of mRNA expression NLRP3, caspase-1, and IL-1b at Figure 1A. Therefore, we add the description as follows, “stimulation with MSU crystals markedly induced mRNA expression of NLRP3, caspase-1, and IL-1b in dose-dependent manner (Figure 1A).”.
- In Figure 2, measurement of PPAR-γ activity was unknown. Additionally, dose-dependent effect of each inhibitor is required.
Answer) Thanks for your thoughtful comment. We detailed the method of PPAR-g activity measurement from nuclear extracts as in No. 8 of your reviewer’s comments. In addition, PPAR-g activities were measured by dose-dependent stimulation of each statin (atorvastatin, simvastatin, and mevastatin) and its result is added at figure 2E. We add the result as follows, “In addition, we found that stimulation with 10mM of atorvastatin, 10mM of simvastatin, or 5mM and 10mM of mevastatin significantly enhanced PPAR-g activity (Figure 2E).”.
- In Figure 3, THP-1 cells transfected with or without PPAR-γ siRNA need to show in clear. Same as for that in Figure 4.
Answer) Thanks for your kind comment. The results shown at figure 3 are described in more detail, as follows “Macrophages transfected with PPAR-g siRNA markedly induced MSU crystal–induced caspase-1 and IL-1b mRNA expression, compared with cells transfected with NC siRNA but not NLRP3 mRNA (Figure 3A). Furthermore, in cells transfected with PPAR-g siRNA, all HMG-CoA reductase inhibitors including atorvastatin, simvastatin, and mevastatin significantly augmented NLPR3, caspase-1, and IL-1b mRNA expression, compared with non-transfected cells.”.
- In Figure 5, NLRP3 inflammasome activation through deactivation of caspase-1 that was ignored in the figure. Why?
Answer) Thanks for your kind comment. We identified that NLPR3 inflammasome activation was inhibited through deactivation of caspase-1 by PPAR-g. Therefore, we insert caspase-1 below NLRP3 inflammasome in figure 5.
- Did antioxidant show same effects as Statins?
Answer) Thanks for your valuable comment. In response to your question, although simple, an experiment using two representative antioxidants is conducted. As a result, the mRNA expression of PPAR-g, NLRP3, caspase-1, and IL-1b by MSU crystals is similar to those of statins. It showed similar results to the anti-inflammatory effect of statins by regulating the PPAR-ROS pathway in this study. The figures illustrated below are the result of additional experiment. However, please note that these results are not officially presented in this study.
- PPAR-γ activity was measured by an assay kit (Abcam) that needs reference(s) to support.
Answer) Thanks for your comment. PPAR-g transcription factor assay kit ab133101, which is a non-radioactive, sensitive method for detecting specific transcription factor DNA binding activity in nuclear extracts, is used in this experiment. We add new reference (No. 37) using PPAR-g transcription factor assay kit manufactured by Abcam.
Reference No. 37) Shou, X.; Zhou, R.; Zhu, L.; Ren, A.; Wang, L.; Wang, Y.; Zhou, J.; Liu, X.; Wang, B.; Emodin, A. Chinese Herbal Medicine, Inhibits Reoxygenation-Induced Injury in Cultured Human Aortic Endothelial Cells by Regulating the Peroxisome Proliferator-Activated Receptor-γ (PPAR-γ) and Endothelial Nitric Oxide Synthase (eNOS) Signaling Pathway. Med. Sci. Monit. 2018, 24, 643-651.
- Variations between Statins were not conducted. Why?
Answer) Thanks for your comment. The reason why the difference of the statins’ effect was not verified in this study was that, as shown in the results, there was no significant differences in inhibiting or increasing the target genes among experimental statins. As mentioned in the limitations of this study, this is presumed to be related to the fact that all of the experimental drugs are lipophilic.
